# Liver Injury and the Macrophage Issue: Molecular and Mechanistic Facts and Their Clinical Relevance

**DOI:** 10.3390/ijms22147249

**Published:** 2021-07-06

**Authors:** Siyer Roohani, Frank Tacke

**Affiliations:** Department of Hepatology and Gastroenterology, Campus Virchow-Klinikum and Campus Charité Mitte, Charité University Medicine Berlin, 13353 Berlin, Germany; Siyer.Roohani@charite.de

**Keywords:** hepatic macrophages, Kupffer cells, monocyte-derived macrophages, acute liver injury, chronic liver injury, NAFLD, NASH, liver fibrosis, HBV, HCV

## Abstract

The liver is an essential immunological organ due to its gatekeeper position to bypassing antigens from the intestinal blood flow and microbial products from the intestinal commensals. The tissue-resident liver macrophages, termed Kupffer cells, represent key phagocytes that closely interact with local parenchymal, interstitial and other immunological cells in the liver to maintain homeostasis and tolerance against harmless antigens. Upon liver injury, the pool of hepatic macrophages expands dramatically by infiltrating bone marrow-/monocyte-derived macrophages. The interplay of the injured microenvironment and altered macrophage pool skews the subsequent course of liver injuries. It may range from complete recovery to chronic inflammation, fibrosis, cirrhosis and eventually hepatocellular cancer. This review summarizes current knowledge on the classification and role of hepatic macrophages in the healthy and injured liver.

## 1. Introduction

Apart from its central role as a metabolic organ, the liver is of critical importance for immune homeostasis and immunological responses to injury. It is located at an intersection of systemic circulation receiving arterial blood through the hepatic artery and portal venous gut-derived blood, enriched with nutrients and microbial products. This gatekeeper position gives the liver not only the challenging task to extract and adapt to innocuous nutrient antigens, but also to identify, bind, inactivate and filter potentially harmful antigens before entering the systemic circulation. To fulfill these tasks the liver harbors the largest population of tissue macrophages in the human body, the so-called Kupffer cells [1,2,3]. Hepatic macrophages consist of tissue-resident Kupffer cells and bone marrow-derived monocytes migrating from the blood stream into the liver tissue, giving rise to bone marrow-derived macrophages [4]. Within the fine microanatomical meshwork of sinusoidal vessels supplying each hepatocyte (the functional, metabolic unit in the liver) Kupffer cells are commonly situated within two branching sinusoids in the periportal area. Equipped with long cytoplasmic protrusions located close to the entrance of supplying blood vessels, these immobile, self-sustaining and locally proliferating macrophages specialize in clearance and gatekeeper function to incoming antigens [5]. Interestingly, most recent studies revealed the asymmetric localization of immune cells in the liver to be orchestrated by liver sinusoidal endothelial cells (LSECs) and adapted to incoming commensal bacteria. LSECs sense commensal bacteria, adjust the pericellular matrix and generate chemokine gradients for immune cells to form zones in order to optimize host defense [6].

In contrast to Kupffer cells, bone marrow-derived macrophages account for only a minor proportion of the hepatic macrophage pool in the healthy liver; they are mainly located at the portal triad [1]. Principally, Kupffer cells are considered tolerogenic phagocytes while bone marrow-derived, infiltrating macrophages represent an inflammatory phenotype. However, the functional phenotype and differentiation is highly diverse and dependent on the interaction between these phagocytes and their microenvironment. Therefore, the traditionally applied model of M1 (inflammatory) and M2 (anti-inflammatory) macrophages [7], which is based on “artificially” induced polarization states in vitro, does not adequately reflect the complex heterogeneity of macrophage polarization in the liver in vivo [8].

Upon liver injury, bone marrow-derived monocytes rapidly infiltrate the liver tissue and differentiate into macrophages, thereby largely augmenting the hepatic macrophage pool. In damaged liver, the microenvironment alters radically. Inflammatory mediators are released by activated and stressed cells while the number of dead cells increases. These microenvironmental changes have a tremendous impact on the phenotypes of Kupffer cells and monocyte-derived macrophages. Their phenotypes determine the functional contribution of macrophages to tissue restoration or aggravation of liver injury [9,10,11].

In this review, we will update the current knowledge on subsets of hepatic macrophages during homeostasis and their role in different liver injury entities. We will also discuss potential implications for ameliorating liver injury in the clinical setting.

## 2. Hepatic Macrophages in Homeostasis

Under homeostatic conditions Kupffer cells, named after the German anatomist Karl Wilhelm von Kupffer (1829–1902) [12], are the predominate macrophage population in the liver. The initial notion that these non-migratory, tissue-resident, and self-sustaining phagocytes embryonically originate from hematopoietic stem cells [13] was recently challenged. Current data suggests that Kupffer cells derive from yolk sac erythromyeloid progenitors that colonize the fetal liver in embryonic day 8.5 in mice. At embryonic day 9.5 these cells give rise to macrophage precursors via signaling of the CX_3_C chemokine receptor 1 (CX_3_CR-1). Through upregulation of the transcription factor inhibitor of DNA binding 3 (ID3) the macrophage precursors eventually develop into Kupffer cells [14]. The half-life of a Kupffer cell is 12.4 days in mice, replenishment occurs through self-renewal which is tightly regulated by the transcription factors MafB and c-Maf [15]. Nonetheless, in case of Kupffer cell death or depletion, as it occurs in acute liver injury, circulating monocytes can contribute to the Kupffer cell pool as well [16] (Figure 1). This adaptability suggests that monocytes as hematopoietic derivates may develop into Kupffer Cells despite their distinct embryonic origin [17,18].

Kupffer cells have numerous essential functions for tissue integrity far beyond being highly effective phagocytes. Apart from the recognition, ingestion and processing of foreign material and cellular debris [19], Kupffer cells play a key role in antimicrobial defense [20]. Equally important is their mediator function in maintaining tolerance to innocuous antigens that continuously pass through the liver before entering the systemic circulation [9].

Moreover, Kupffer cells participate in important metabolic pathways. They recycle iron by phagocytosing damaged red blood cells (RBC) and metabolizing heme-derived, histotoxic free iron [21,22,23]. Recent evidence suggests that hepatic iron accumulation, particularly heavily iron-loaded Kupffer cells, are involved in the pathogenesis of nonalcoholic steatohepatitis (NASH) and liver fibrosis [24]. Kupffer cells are also involved in the degradation of aged platelets, as the depletion of Kupffer cells has prevented the removal of aged platelets on intravital imaging. Thereby, the calcium-dependent c-type lectin named macrophage galactose lectin (MGL) receptor on macrophages [25] interacts with desialylated, aged and potentially dysfunctional platelets [26].

Additionally, resting Kupffer cells are the main source of the plasma level of cholesteryl ester transfer protein (CETP) [27]. This pore-forming protein enables the transfer of cholesterol from the antiatherogenic, anti-inflammatory high-density lipoproteins (HDL) to the atherogenic low (LDL) and very low-density lipoproteins (VLDL), that deliver cholesterol to the peripheral tissue [28]. Interestingly, lipopolysaccharides (LPS) from Gram-negative bacteria induce Kupffer cell activation, which significantly reduces CETP synthesis, thereby raising HDL levels. These findings may imply that HDL is involved in host defense and that CETP and HDL signaling does not only correlate but is mechanistically connected to liver inflammation [29].

Kupffer cells are characterized by the expression of the surface markers F4/80, CD11b^+/low^, CD68 and, in mice, C-type lectin domain family 4 member F (CLEC4F) [10]. To distinguish Kupffer cells from bone marrow-derived macrophages, the expression of T cell immunoglobulin (Ig), mucin domain containing 4 (Timd4), stabilin 2 (Stab2) gene receptors and the paucity of CX_3_CR1 are being used as well [9,11]. Comprehensive studies of hepatic transcriptomes using single-cell RNA-sequencing of human samples have broadened our view on the heterogeneity of macrophages in the healthy human liver. While CD68 is a known marker for human and mouse liver macrophages [10], the authors subdivided the macrophages in two populations, MARCO^+^ and MARCO^−^ (**MA**crophage **R**eceptor with **CO**llagenous structure) as well as CD14^+^ monocytes [30,31]. Of note, transcriptional profiles of MARCO^+^ cells demonstrate an anti-inflammatory, tolerogenic phenotype. One example to mention is the transcription of V-set and immunoglobulin domain-containing 4, also known as Complement Receptor of Immunoglobulin superfamily (VSIG4, CRIg), a tolerogenic co-signaling molecule inhibiting T-cell immune responses [32]. Combined with their periportal location, these cells appear similar to long-lived, sessile, liver resident Kupffer cells identified in mice [10,11,30]. In contrast, MARCO^−^ macrophages display a pro-inflammatory transcriptional profile with decreased expression of CD 163, an anti-inflammatory scavenger receptor for hemoglobin-haptoglobin complexes that prevents intravascular oxidative stress damage from free hemoglobin, as it occurs in pathological conditions of intravascular hemolysis (malaria, hemoglobinopathies, autoimmune hemolysis, and drug-induced hemolysis) [33]. This phenotype of MARCO^−^ macrophages is similar to recently recruited pro-inflammatory macrophages [10].

To fulfill their diverse tasks, Kupffer cells are versatilely equipped with complement, antibody, scavenger and pattern recognition receptors (e.g., toll-like receptors (TLR)) [34]. TLRs recognize pathogen-associated molecular patterns (PAMPs) such as LPS, consequently mounting an immune response to effectively neutralize invading microorganisms [20]. Furthermore, TLR activation on Kupffer cells induces immunogenic T cell responses [9]. Another essential and unique receptor for Kupffer cell mediated host defense is CRIg. CRIg binds the complement factors and opsonins C3b and iC3b (inactivated C3b), which allows them to phagocytose pathogens [35]. Elaborate intravital imaging revealed CRIg’s ability to directly bind and capture lipoteichoic acid (LTA) from *Staphylococcus aureus* (Gram-positive bacterium) out of the blood flow and independent of complement factors acting as a pattern recognition receptor (PRR) [36]. These adapted properties corroborate the role of Kupffer cells and the liver as a guardian for the body’s host defense [20].

As mediators of immune tolerance, Kupffer cells orchestrate antigen uptake, processing and presentation to induce regulatory T cells and maintain tolerance [9,37]. Simultaneously, Kupffer cells suppress T cell activation from other antigen presenting cells without the need for immunosuppressive cytokines like interleukin-10 (IL-10), transforming growth factor β (TGF-β) and nitric oxide (NO) [38,39].

## 3. Hepatic Macrophages in Acute Liver Injury

Acute liver injury is defined clinically as a (more than) two- to threefold increase of liver transaminases above the upper limit of normal (marker of hepatocyte damage), jaundice and impaired coagulation function (International Normalized Ratio (INR) > 1.5) of hepatic origin in a patient without preexisting chronic liver disease. Acute liver failure additionally comprises altered mentation (termed hepatic encephalopathy) and is initiated by a severe acute liver injury [40], a life-threatening condition. The most common causes include hepatotoxic drugs (e.g., acetaminophen/paracetamol, phenprocoumon, antibiotics, antiepileptics), herbal or dietary supplements and acute viral hepatitis (hepatitis A (HAV), B (HBV) and E (HEV) viruses), although a multicenter data analysis from Germany revealed that in about one quarter of cases the cause remains unknown. Approximately one third to half of all patients with acute liver failure require emergency liver transplantation as a last resort [41,42].

The immense injury to the liver tissue greatly alters both, the microenvironment and immune cells. Danger-associated molecular patterns (DAMPs), a group of endogenous danger molecules released from damaged or dying cells such as high mobility group box 1 (HMGB1) or mitochondrial DNA (mtDNA) [43] are released and bind to pattern recognition receptors (PRR) such as TLRs on Kupffer cells, thereby activating them (Figure 2) [44]. The recognition of these danger signals leads to the formation of the inflammasome.

The inflammasome is a family of multicomponent protein complexes in the cytosol of macrophages as well as other hepatic cells (e.g., hepatic stellate cells, hepatocytes) and part of the innate immune system. It consists of a NOD-like receptor (NLR), an adaptor protein *apoptosis-Associated speck-Like protein containing a caspase recruitment domain* (ASC) that links two other components together, an amino-terminal pyrin domain and a carboxy-terminal caspase recruitment domain (CARD), and the proenzyme pro-caspase 1. Inflammasome activation eventually leads to the secretion of interleukin-1 (IL-1β) and interleukin-18 (IL-18) [45]. The activation requires two essential signals. The first signal induces the transcription of inflammasome components and cytokines for the inflammatory response while the second signal triggers their release. In the first step pattern recognition receptors, for example, TLRs become activated by PAMPs or DAMPs such as heat shock proteins (HSPs), mt-DNA and nuclear DNA, all of which indicate cell damage. The pattern recognition receptors are coupled to the adaptor protein MyD88. Through interaction with the transcription factor Nuclear Factor-κB (NF-κB) the expression of inflammasome proteins and cytokines are upregulated [46,47]. The second signal is initiated through the activation of the NLR by various stimuli such as free ATP, K^+^ ionophores or heme. Numerous NLR proteins may be involved in inflammation activation, however *Nucleotide-binding oligomerization domain, Leucine rich Repeat and Pyrin domain containing* (NLRP3) is the most thoroughly studied. After binding to its pattern recognition receptors, the pyrin domain of NLRP3 aggregates with the adaptor protein ASC. The CARD domain of ASC then interacts with CARD domains from procaspase-1 thereby inducing an autocleavage of procaspase-1 to form the active caspase-1 enzyme [46,47,48]. Caspase-1 consequently activates IL-1β, IL-18 and a cytosolic protein gasdermin D [49,50]. The latter facilitates the release of IL-1β and IL-18 from the cell by forming pores in the plasma membrane [51]. Consequently, intracellular compounds are released acting as DAMPs, thereby inducing inflammation and cell death, a process termed pyroptosis. Mechanistically, the cell defends itself through pyroptosis against intracellular pathogens by exposing them to immune factors and eradicating their intracellular replication niche [52,53,54]. Simultaneously, IL-1β is released and has pleiotropic inflammatory effects; it affects the expression of fever regulating proteins, reduces vascular tone and increases vascular permeability of infiltrating immune cells. IL-18 is essential for interferon-γ release and contributes to adaptive immunity by promoting T helper 2 subset cell-mediated Interleukin-4 (IL-4) release [55]. The inflammasome-mediated cytokine release stimulates immune cells, particularly neutrophils and macrophages, thus initiating an immune response towards damaged tissue [56,57]. The inflammasome is involved in various entities of liver injury such as viral hepatitis, bacterial infection, alcoholic liver disease (ALD) [58,59,60] and NASH. In animal models fed a NASH-inducing diet the blockade of the NLRP3 inflammasome with the small molecule MCC950 visually improved fibrosis and reduced the expression of inflammasome components (caspase 1, IL-1β) and markers of cell damage (AST/ALT) [61].

Once activated, Kupffer cells crosstalk with local and remote immune cells by secreting a wide range of cytokines [34]. The release of chemokine ligand 2 (CCL2) from Kupffer cells and other hepatic cells (hepatocytes, stellate cells) leads to the recruitment of circulating monocytes expressing inflammatory chemokine receptors (like CCR2) [62]. Additional chemokines that attract bone marrow-derived monocytes are CCL1, CCL25 and CX_3_CL1 (the latter also known as fractalkine) [63,64,65]. Moreover, the medium-chain fatty acid receptor GPR84 draws monocyte-derived macrophages and neutrophils to the site of acute liver injury, as antagonization with highly specific small molecule GPR84 inhibitors significantly reduced infiltration rates in mice [66].

The hepatic macrophage pool is rapidly expanded by infiltrating monocytes. Shortly thereafter, monocyte-derived macrophages represent the main macrophage population [11,62]. The dynamic changes continue as the infiltrating macrophages do not represent a homogenous group, but rather consist of distinct subpopulations with different, convertible phenotypes [67]. In murine models, Ly6C^hi^ macrophages display a pro-inflammatory phenotype with expression of PRRs and chemokine receptors (e.g., CCR2), while Ly6C^low^ macrophages patrol through the liver and show restorative pro-resolution properties [9,18,62,68]. They express matrix metalloproteinases (MMPs) termed MMP9, MMP12 and MMP13, a group of enzymes that split peptide bonds from proteins in the ECM, a crucial step in various processes like wound healing, cell proliferation, migration differentiation and angiogenesis [62,67,69,70]. Interestingly, recently infiltrated Ly6C^hi^ macrophages can transform into Ly6C^low^ phagocytes at the site of liver injury, which is promoted by macrophage colony-stimulating factor 1 (CSF-1) [71]. In a model of focal sterile injury, macrophages initially reside along the site of injury for at least 48 h forming a ringlike structure. Then they downregulate chemotactic receptors like CCR2 while CX_3_CR1 receptors are upregulated. After this phenotypic conversion, macrophages enter the site of injury [72]. The converted Ly6C^low^ macrophages promote tissue repair by inducing angiogenesis via vascular endothelial growth factor A (VEGF-A), reconstructing the extracellular compartment, phagocytosing and disposing of dead cells [11,71].

In rodent models, the timepoint of inhibiting bone marrow-derived macrophage infiltration tremendously influences the prognosis of the liver tissue ranging from complete restoration to fibrosis and chronification of liver injuries. In an animal model of acetaminophen (APAP)-induced acute liver injury the prevention of bone marrow-derived macrophage infiltration through CCR-2 antagonization ameliorated liver injury [62]. However, the subsequent inflammation resolution and dead cell clearance mediated by converted Ly6C^hi^ was impaired as well [73]. This supports the idea of phenotype switching macrophages from pro-inflammatory Ly6C^hi^ to restorative Ly6C^low^ phagocytes and corroborates their orchestrating role in tissue restoration and inflammation. Importantly, other immune cells capable of phagocytosis, like neutrophils but also platelets, remarkably affect the course of acute liver injuries. Blockade and genetic deletion of CLEC-2 (C-type-lectin-like-2) receptors on platelets leads to increased TNF-α release from liver macrophages which consequently activates neutrophils that rapidly clear the site of injury from debris and support tissue restoration [74]. Moreover, the platelet adhesion molecule von Willebrand factor (VWF) increases during acute APAP-induced liver injury which promotes platelet aggregation and delays tissue repair. Accordingly, pharmacological antagonization or genetic deletion significantly accelerated tissue repair [75].

A different reservoir of fully mature F4/80^+^ CCR2^−^ CX_3_CR1^−^ macrophages expressing the transcription factor GATA6, named after the DNA sequence *guanine-adenine-thymine-adenine* the transcription factor binds to, has been identified in the peritoneal cavity. These phagocytes rapidly cross the mesothelium and use CD44 to infiltrate the subcapsular areas of the liver in about one hour after sterile liver injury in mice. Sterile liver injury is defined as an inflammatory response due to tissue injury without infection [76]. The macrophages were drawn to the site of injury by ATP from necrotic cells and show a reparative phenotype as the knockout of GATA6 decreased the number of visible macrophages at the site of injury and delayed tissue repair. Intriguingly, these cells were not able to pass through blood vessels (extravasation) if placed in the blood stream before entering the site of injury [77]. However, most recent data has demonstrated GATA6 macrophages to also hold migratory and hemostatic properties of platelets in the peritoneal cavity [78]. Since the depth of focus of spinning disc intravital microscopy applied in the studies that identified GATA6 peritoneal macrophages as responders to liver injury are limited, it remains to be elucidated, whether these mechanisms also occur in deeper liver tissue more distant to the peritoneal cavity. Another limiting factor in this matter was demonstrated in recent studies, where cavity macrophages downregulate GATA6 upon contact with injured tissue, which complicates the tracing of their fate in vivo [79].

## 4. Chronic Liver Injury

### 4.1. Hepatic Macrophages in Nonalcoholic and Alcoholic Fatty Liver Disease

All forms of chronic liver injury, such as nonalcoholic fatty liver disease (NAFLD), alcoholic liver disease (ALD) or chronic viral hepatitis B and C are driven by continuous hepatic inflammation and lead to fibrosis as a uniform wound healing response. If the inflammatory status does not subside, fibrosis eventually leads to cirrhosis and hepatocellular carcinoma [1,80].

Nonalcoholic fatty liver disease is the most common liver disease affecting around 24% of the population worldwide and represents the liver manifestation of obesity and metabolic syndrome [81]. Through tremendous advances in the treatment of viral hepatitis B and C on the one hand and the ever-increasing global burden of obesity even in younger generations on the other hand, NAFLD and its sequelae are estimated to become the most common causes of liver cirrhosis and hepatocellular carcinoma [82,83,84].

NAFLD is an umbrella term for a spectrum of liver diseases in a patient with no other causes for secondary hepatic fat accumulation and particularly with no significant alcohol intake (<30 g/d for men, <20 g/d for women) [85]. The typical pathophysiological sequence starts with hepatic steatosis, a histological accumulation of triglycerides in >5% of hepatocytes, to nonalcoholic steatohepatitis, a chronic inflammatory condition with necrotic hepatocyte injury leading to NASH-fibrosis, cirrhosis and eventually hepatocellular carcinoma [85]. Yet, this process may not be considered linear as steatosis may also directly lead to fibrosis or inflammation resolution may improve and restore tissue integrity. The degree of fibrosis is defined histologically by the number of fibrotic septa from F0 (no fibrosis) over F1 (portal fibrosis without septa), F2 (portal fibrosis with few septa), F3 (numerous septa, “bridge building fibrosis”) to F4 being cirrhosis. The fibrotic stage is a predictor for liver-related morbidity, liver-related mortality and overall mortality [86,87]. In noncirrhotic NAFLD fibrosis (F0-F3 fibrosis) the main causes of death in NAFLD patients are cardiovascular diseases and non-hepatic malignancies [88]. However, the liver-related mortality exponentially grows and supersedes all other causes of mortality once a cirrhotic liver condition (F4 fibrosis) is established [89]. Nevertheless, NAFLD has to be considered the hepatic manifestation of a systemic burden which emphasizes the necessity for interdisciplinary care of patients with NAFLD [90].

The fat overload in NAFLD and ALD causes cell death of hepatocytes and the release of DAMPs that activate macrophages and their inflammasomes [91,92]. Another inflammatory pathway was described in in vitro studies of NASH where fatty acids (palmitate or lysophosphatidyl-choline) were able to induce the release of extracellular vesicles containing TNF-α-related apoptosis-inducing ligands (TRAIL) from hepatocytes. The TRAILs increased the transcription of IL-1β and IL-6 in monocyte-derived macrophages from mice thereby promoting inflammation [93]. Furthermore, the liver-derived plasma protein histidine-rich glycoprotein (HRP) has been shown to skew macrophages towards inflammatory phenotypes in mouse models of NASH and tumors [94,95]. The glucocorticoid-induced leucine zipper (GILZ), a multifunctional protein that spreads the anti-inflammatory signaling of glucocorticoids in numerous tissues, is downregulated in Kupffer cells of murine models of hepatic steatosis [96]. Consequently, Kupffer cells displayed an inflammatory phenotype with elevated expression of CCL-2, TNF-α and IL-6, which was improved in GILZ overexpressing transgenic mice [97].

Recent studies using elaborative tools for lineage tracing, like bone marrow chimeras and parabiosis, have discovered a decreasing number of self-renewing Kupffer cells in experimental NASH models. Replenishment of the hepatic macrophage pool was carried out by bone marrow-derived Ly6C^+^ monocytes giving rise to Kupffer cells. The infiltrated cells exhibited more pro-inflammatory characteristics than embryonic resident Kupffer cells and remain in the tissue after disease regression. Interestingly, the infiltrated Kupffer cells have less lipid storage capacity than resident Kupffer cells [98].

In line with the notion of macrophage heterogeneity in NASH models, another comprehensive study applying unbiased single cell transcriptome analysis in mice revealed and confirmed: (i) resident Kupffer cells are gradually lost and replaced by monocyte-derived Kupffer cells during disease progression; (ii) resident Kupffer cells are not pro-inflammatory, suggesting that inflammatory phenotypes in previous studies might have been associated with infiltrating rather than resident Kupffer cells; (iii) infiltrating monocytes differentiated into another CLEC4F^−^ osteopontin-expressing macrophage subpopulation with a transcriptomic profile similar to lipid associated macrophages (LAMs) [99]. Supporting this idea, another RNA sequencing study of human hepatic macrophages from lean individuals and obese, insulin resistant patients have shown no differences in the transcription of inflammatory markers. An interesting secondary finding was made using interfering RNA (iRNA) to inhibit RELA expression, a protein essential for NF-κB processing. The inhibition of RELA did not affect the transcription of inflammatory cytokines suggesting an alternative inflammatory pathway apart from NF-κB in hepatic macrophages [100]. Most recently, a remarkable contribution of XCR1+ type 1 conventional dendritic cells (cDC) to the hepatic myeloid cell pool has been described in NASH patients and murine NASH models. Using single-cell transcriptome analysis, the authors found rapidly recruited cDC progenitors deriving from the bone marrow. The proliferating cDCs quickly outnumber Kupffer cells and promote inflammatory T-cell reprogramming which adversely influences NASH phenotypes [101].

Another remarkable finding on macrophage heterogeneity in NASH has discovered specialized macrophages in patients and murine models, both of which highly express CD9 and Triggering Receptor Expressed on Myeloid Cells 2 (TREM-2), a scavenger receptor involved in apoptotic cell clearance [102,103]. These cells were termed scar-associated macrophages or lipid-associated macrophages as they showed close spatial relation to the fibrotic niche. However, a clear functional correlate is missing to date, strengthening the need to correlate single-cell transcriptomics with functional studies. Strikingly, these TREM-2+ scar-associated macrophages share many similarities with “lipid-associated macrophages” in inflamed adipose tissue in obesity models, in which these cells controlled adipose tissue characteristics [104]. Further evidence suggests that TREM-2 actually has a regulatory role in various rodent liver injury models [105,106]. However, these TREM-2 positive macrophages could be a heterogeneous subset with opposing functions depending on the context of liver injury.

### 4.2. Hepatic Macrophages in Viral Hepatitis B and C

Chronic viral hepatitis caused by HBV or HCV remain a meaningful cause of liver associated morbidity and mortality [107]. The research on immunocompetent animal models is limited by the number of appropriate models as, for example, mice have a natural immunity against HCV and research on chimpanzees is hampered by financial and ethical constraints [108].

Similar to their multifaceted tasks in other liver diseases, hepatic macrophages can contribute to antiviral responses upon HBV or HCV infection. Research on the hepatotropic lymphocytic chroriomeningitis virus (LCMV) in mice has shown that the liver rapidly recruits pro-inflammatory macrophages (within 24 h) to support local Kupffer cells, when acutely infected [109]. In vitro cell culture studies of human Kupffer cells exposed to HBV surface antigen (HBsAg) have revealed increasing productions of the inflammatory cytokines TNF-α, IL-6 and CXCL8 (IL-8) that peaked after six hours of exposure [110]. Through NF-κB mediated transcription, these inflammatory cytokines, most importantly, IL-6 prohibit viral spreading in infected hepatocytes. IL-6 activates the mitogen-activated protein kinases exogenous signal-regulated kinase 1/2 (ERK1/2), and c-jun N-terminal kinase (JNK), two members of the mitogen-activated protein kinases (MAPKs) that transfer extracellular stimuli to a wide range of cellular stimuli [111]. ERK1/2 and JNK inhibit expression of hepatocyte nuclear factor (HNF) 1α and HNF4α, two transcription factors essential for HBV gene expression and replication [112]. Similarly, human Kupffer cells and monocyte-derived macrophages incubated with HCV in vitro activated the inflammasome and NF-κB via TLR2, which induced IL-1β and IL-18 secretion [113,114]. In line with the aforementioned two signal activation of inflammasomes, HCV exposed human macrophages showed: (i) viral RNA triggers MyD88-mediated TLR7 signaling to induce IL-1β mRNA expression; (ii) HCV uptake concomitantly induces a potassium efflux that activates the NLRP3 inflammasome for IL-1β processing and secretion; (iii) HCV infection is directly linked to liver inflammation by NLRP3 inflammasome activation [115].

Moreover, the depletion of Kupffer cells via clodronate-liposomes, a specific bisphosphonate that depletes macrophages [116], led to a rapid dissemination of LCMV in mice due to the inability to capture and process viral particles [117]. It is therefore appropriate to consider Kupffer cells a critical immune barrier in acute HBV and HCV infections.

Nevertheless, during chronic phases of viral hepatitis, hepatic macrophages can also diminish immune responses and even support viral persistence [118]. Chronic HBV and HCV infection leads to Kupffer cell-mediated release of immunomodulatory cytokines, such as IL-10, TGF-β, PD-L1 and PD-L2 that suppress T cell responses and promote viral persistence [119,120,121]. Studies on human samples confirmed the pro-chronification and immunosuppressive phenotypes of hepatic macrophages in patients with chronic HBV infections [122]. In murine models of chronic HBV using intravenous injections of engineered, replication-competent HBV plasmids, Kupffer cells reproducibly induced tolerance to HBV by producing IL-10 that persisted for up to 3 months after a single injection [123,124]. In the same animal model TLR2 was identified as a causative receptor that led to IL-10 induced CD8+ T cell exhaustion by which HBV clearance was hindered [125]. The injection of HBV plasmids was also used to study chronic liver injury by HBV infections in offspring of infected mothers. Strikingly, Kupffer cells in the offspring induced tolerance through PD-L1 by which cytotoxic CD8+ T cell mediated lysis of infected cells was inhibited and HBV persistence was enabled [126].

### 4.3. Fibrosis and Cirrhosis Modulation by Hepatic Macrophages

Fibrosis describes excessive scarring overproportionate to a wound healing response towards tissue injury [127]. If left untreated, progressive hepatic fibrosis leads to cirrhosis, a histoarchitectural remodeling with abundant collagen deposition that becomes clinically overt by decreasing hepatic function [128]. The chronic inflammation predisposes to carcinogenesis while increasing portal pressure triggers numerous clinical complications with significant morbidity and mortality [129]. Both, clinical observations and experimental models over the last years, challenged the former belief of irreversible liver fibrosis and identified new targets to halt and reverse this process [130].

Hepatic stellate cells (HSCs) transition to myofibroblasts and represent the main matrix-producing cells in the liver. Activation is induced by direct cell–cell interaction or binding of fibrogenic mediators [131].

In the dynamic process of inflammation and fibrogenesis, liver macrophages hold a dual function; during fibrogenesis the depletion of hepatic macrophages in mice improves scarring, whereas the depletion during resolution phases impedes adequate tissue restoration pointing towards functionally distinct subpopulations of macrophages within this process [132]. Upon damage to the hepatic microenvironment DAMPs and PAMPs are released that trigger local non-parenchymal cells (Kupffer cells, hepatic stellate cells, liver sinusoidal endothelial cells) to release a broad variety of inflammatory and profibrogenic soluble mediators. These mediators (e.g., CCL2) attract inflammatory immune cells like LyC6^hi^ bone marrow-derived macrophages and activate matrix-producing profibrotic cell populations to form scar tissue [34,127]. Besides the recruitment of inflammatory and profibrotic macrophages by CCL2, Kupffer cells can also directly promote activation and survival of HSCs and myofibroblasts through secretion of growth factors (PDGF, TGF-β) and CCL5 [133,134,135,136,137].

The interaction of Kupffer cells with other immune cells, such as natural killer T (NKT) cells, can also promote their profibrotic phenotype. Kupffer cell derived CXCL16 recruits profibrotic NKT cells through its ligand receptor CXCR6 to the site of liver injury [138].

In fibrogenesis and chronic liver injury, monocyte-derived macrophages display a similar functional switching from inflammatory/fibrogenic to pro-resolutive/antifibrogenic as described earlier in acute liver injury. Several observations from mouse models of hepatic fibrosis and its resolution support the phenotypic adaptation of macrophages. The selective depletion of early infiltrating Ly6C^hi^ macrophages reduces HSC activation and extracellular matrix (ECM) formation, while depletion of Ly6C^low^ macrophages during regression phases compromises ECM breakdown which preserves fibrosis [67,132]. Pharmacological blockade of Ly6C^hi^ infiltration using sophisticated Spiegelmer-based CCL-2 antagonists (artificial oligonucleotides specifically binding CCL-2), named mNOX-E36, augmented the proportion of pro-restorative Ly6C^low^ macrophages and accelerated fibrosis regression in animal models of chronic liver disease [139]. Mechanistically, inflammatory Ly6C^hi^ monocyte-derived macrophages use TGF-β and IL-13 to activate HSCs [67,136,140,141,142]. The phenotype switching from Ly6C^hi^ to Ly6C^low^ macrophages occurs after phagocytosis of dead cells (efferocytosis). The responsible signaling pathway comprises the receptor and tyrosine kinase Janus kinase (JAK) and signal transducer and activator of transcription (STAT) DNA-binding proteins. They mediate the signaling and downstream biological effects in response to binding of IL-10 and IL-6 [69,143,144]. Additional macrophage receptors for efferocytosis and consequent phenotype switching are PtdSer-dependent receptor tyrosine kinases (RTKs) AXL and the proto-oncogene tyrosine-protein kinase MER (MERTK). Both are activated by IL-4 or IL-13 and lead to induction of anti-inflammatory and tissue repair responses in macrophages [145].

A comprehensive study of more than 100,000 single human cells identified scar-associated TREM2^+^ CD9^+^ subpopulation of macrophages that derive from circulating monocytes. Furthermore, the group discovered a number of pathways involved in fibrogenesis such as NFRSF12A, PDGFR and NOTCH signaling [103]. Supporting these findings, another study on epigenetic changes in mouse models of NASH confirmed the existence of TREM2^+^ CD9^+^ macrophages (corresponding to scar-associated macrophages in humans) localized in the fibrotic niche. Moreover, the liver X receptor (LXR), a nuclear receptor for identity and survival of Kupffer cells, is epigenetically modified in NASH models, which promotes a scar-associated macrophages phenotype [146].

In summary, DAMPS and PAMPs released upon liver injury immediately trigger local Kupffer cells to alarm and recruit monocytes from the blood stream into the tissue to support the inflammatory situation. Initially, macrophages intend to defend and neutralize against potential noxa (e.g., toxins, invading microorganisms, etc.) and encourage damage control through collagen synthesis. After this initial alarming inflammatory/defense and damage control situation, macrophages biologically seem to notice the need for tissue repair, disposal of debris, collagen breakdown and restoration of tissue integrity. This dynamic process offers a window of opportunity for preventive or therapeutic interventions.

## 5. Therapeutic Approaches and Conclusions

Numerous target points and approaches with promising results have been identified to clinically tackle liver disease initiation and their final common path of fibrosis and cirrhosis [130,147]. Macrophages represent sentinels of tissue homeostasis and immunological tolerance, are orchestrators in acute liver injury and hold dual, interchangeable functions in liver disease progression. Thus, targeting hepatic macrophages is one auspicious cornerstone of liver disease therapy. Generally, several basic approaches in targeting hepatic macrophages can be considered: (i) prohibit macrophage recruitment; (ii) inhibit macrophage activation; (iii) induce phenotype switching of macrophages. The transfusion of autologous macrophages in diseased patients (cell-based therapy) is another novel approach. In a very small, yet very innovative trial of autologous macrophage transfusion in chronic liver disease patients the safety and feasibility has been proven. An update on this new immunological branch of chronic liver disease has been nicely reviewed [148,149].

### 5.1. Therapeutic Target: Macrophage Recruitment

The accumulation of inflammatory monocyte derived-macrophages in NASH progressing to fibrosis is one essential pathophysiological finding [150,151,152]. The chemokine induced recruitment by CCL2 and CCL5 can be inhibited by using the CCR2/CCR5 antagonist cenicriviroc. Cenicriviroc has initially been successfully tested as an anti-human immunodeficiency virus (HIV) agent, but demonstrated highly effective antifibrotic activity in animal models of chronic liver injury [153,154,155]. The randomized, double-blind multinational phase 2b trial (CENTAUR trial) on 289 NASH patients receiving cenicriviroc or placebo has shown significant improvements in the degree of histologically verified fibrosis after 1 year of treatment [156]. Although the antifibrotic effect has not been clearly visible in the final data after 2 years, the drug has shown an extraordinary safety profile and was well tolerated [157]. Thus, the larger randomized, placebo-controlled, multinational phase 3 trial (AURORA trial) with approximately 2000 NASH patients was initiated [158]. Unfortunately, the trial was recently terminated after the 1 year interim analysis revealed lack of efficacy [159]. The final data to discuss and search for possible explanations or alternative approaches has not yet been published.

Other anti-inflammatory cell recruitment agents with promising preclinical trials are propagermanium, a CCR2 inhibitor [160], the aforementioned oligonucleotide CCL2 antagonist, mNOX-E36 [62], and maraviroc, another anti-HIV agent which inhibits CCL5 [161]. A different approach to inhibit inflammatory macrophage inhibition is to antagonize the medium-chain fatty acid receptor GPR84 which is upregulated in monocytes upon inflammatory activation. Pharmacological inhibition with small molecules reduces myeloid cell infiltration and ameliorates steatohepatitis and fibrosis in NASH models [66].

### 5.2. Therapeutic Target: Macrophage Activation

As intrasinusoidal sentinels residing close to the portal venous and arterial blood supply routes, macrophages are continuously exposed to DAMPs and PAMPs (e.g., LPS) from the intestinal microbiota. Particularly in chronic liver disease and by toxins (such as alcohol) the intestinal permeability increases and macrophage exposure to DAMPs and PAMPs rises. The liver feedbacks through bile and antibody secretion. This reciprocal interaction establishes the concept of the gut–liver axis [162].

Within this process, the activation of macrophages exposed to DAMPS and PAMPs represents another therapeutic target point for treating liver diseases. More specifically, genetic depletion of TLR4 in LPS-challenged mice leads to reduced HSC activation and chemokine secretion which consequently reduces Kupffer cell activation and fibrogenesis. Mechanistically, TLR4 activation on HSCs downregulates and thereby sensitizes the receptor for the profibrotic cytokine TGF-β (bambi receptor) in a NFκB-dependent signaling pathway [163]. Furthermore, the combination of the TLR4 inhibitor seralexin and rosiglitazone, an agonist of peroxisome proliferator-activated receptors (PPARs, *for further discussion see section on induction of phenotype switching in macrophages below*) have proven effective in mouse models of chronic liver disease and fibrosis [164]. Supporting this pathomechanism, inhibition of the PAMP responsive NLRP3 inflammasome by the small molecule MCC950 reduced the severity of liver inflammation and fibrosis in NASH models. Moreover, Kupffer cell activation by cholesterol crystals in vitro to secrete IL-1β were abolished by MCC950 [61].

### 5.3. Therapeutic Target: Macrophage Function and Polarization

Upon liver injury, the rapid influx of inflammatory, profibrogenic bone marrow-derived macrophages is subsequently followed by a phenotype switch of formerly invading macrophages towards regenerative macrophages. Thus, therapeutic induction of this step may be beneficial for patients with chronic liver disease.

In this context, an innovative branch of therapy is depicted by nanoparticles [165]. Nanoparticles are materials measuring about 1–100 nm, individually designed for targeted therapy [166]. They can be loaded with small drug molecules, proteins, DNA and RNA. Additionally, the release of their content can be favorably customized and, for example, be triggered by lower pH (e.g., in lysosomes) or by light [167,168]. In a set of experiments using sophisticated whole body and intra-organ imaging the pharmacokinetics of different vehicles to target hepatic immune cells have been examined [169]. One straightforward approach is the selective administration of immunosuppressants by vehicles to macrophages. In experimental models of acute and chronic liver injury the administration of dexamethasone-loaded fluorescent-tagged liposomes reduced liver injury and fibrosis. Intriguingly, a small proportion was taken up by T cells and led to a significant depletion of those which may also confound and mediate the anti-inflammatory effects [170]. Another earlier study on liver fibrosis models induced by bile duct ligation has shown almost exclusive dexamethasone liposome uptake in Kupffer cells by coupling dexamethasone to mannosylated albumin. However, the study did not find antifibrotic effects, instead, even accelerated fibrogenesis was seen in vivo [171]. Since inflammatory cytokines like TNF-α are detrimental in chronic liver diseases and the systemic administration carries harmful side effects (systemic bacterial infections) [172], nanoparticles loaded with small interfering RNA (siRNA) were used to selectively inhibit TNF-α released by macrophages. Through conjugation with mannose-modified trimethyl chitosan-cysteine (MTC) the nanoparticles were selectively taken up by macrophages via their mannose receptor. In the applied mouse model of acute liver injury by LPS/D-galactosamine the liver damage and lethality were prevented [173]. Thus, macrophages and other antigen presenting cells (APCs) represent an attractive target for nanomedicine in liver diseases.

Galactin-3, a β-galactoside binding protein mainly produced in macrophages, has numerous regulating effects on inflammation and the innate immune system [174]. One of its pleiotropic effects is the activation of myofibroblasts through TGF-β, a critical step in liver fibrogenesis [175]. Galactin-3 inhibition has shown multiple beneficial effects in mouse models of NASH [176]. However, in a multi-center, double blind, phase 2b trial using the Galactin-3 inhibitor belapectin in patients with NASH cirrhosis, neither portal hypertension, measured by the hepatic venous pressure gradient (HVPG), nor histological fibrosis criteria improved. Interestingly, in a subgroup analysis of patients without esophageal varices at baseline, belapectin significantly reduced HVPG and the reoccurrence of varices after 52 weeks [177]. A phase 1 trial on another galectin-3 inhibitor, GB1211, has recently confirmed safety, tolerability and an appropriate pharmacokinetics and pharmacodynamics profile to be further investigated [178].

PPARs (peroxisome proliferator-activated receptors) are a group of three transcription factors (PPARα, PPARβ/δ and PPARγ) essential for liver homeostasis, lipid metabolism, insulin sensitivity and inflammatory responses [179]. PPAR stimulation prevents fibrogenesis by keeping HSCs in the quiescent state and regulating inflammatory responses [180]. A broad number of studies have proven the beneficial effects of PPAR agonism on NAFLD pathology and fibrogenesis. The effectiveness of inducing the pleiotropic metabolic effects of PPARs for NASH models corroborates the concept of NASH not being a single pathological entity but rather the hepatic manifestation of metabolic syndrome [181]. In animal models of NAFLD the pan-PPAR agonist lanifibranor histologically improved fibrosis, steatosis and inflammation. Additionally, lanifibranor attenuated inflammatory activation of human monocytes and murine bone marrow-derived macrophages stimulated with palmitic acid in vitro [182]. These very promising preclinical results of lanifibranor have been tested in a randomized, placebo-controlled, multicenter, phase 2b trial (NATIVE trial) on 247 NASH patients that reported impressive results on histological improvements of NASH and fibrosis after 6 months of therapy [183].

### 5.4. Conclusions

In summary, hepatic macrophages are a highly versatile population and key players in maintaining homeostasis and immune tolerance. Under various tissue injury conditions, they rapidly expand in numbers and have a two-faced phenotype, encouraging inflammation and fibrosis, yet also supporting restoration and inflammation resolution. Both, preclinical and clinical research over the last years have generated valuable knowledge on cellular heterogeneity and molecular pathways controlling their activation as well as their function. Elaborative single cell analysis and intravital imaging will allow direct and individual analyses of cells in the complex interplay of immune system, local microenvironment, microbiota and blood vessels in different liver diseases culminating in fibrosis, cirrhosis and carcinogenesis.

## Figures and Tables

**Figure 1 ijms-22-07249-f001:**
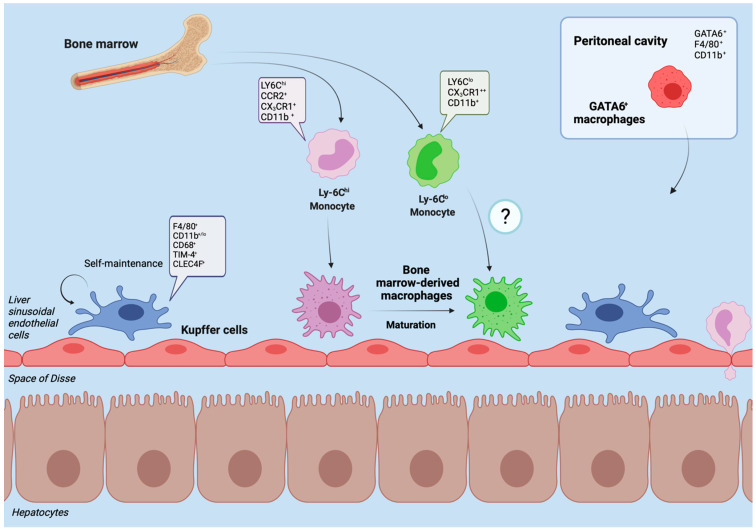
Liver macrophages in homeostasis. The figure displays the heterogenous subpopulations of macrophages in the healthy liver of mouse models. Kupffer cells represent the self-renewing, tissue-resident and dominant phagocyte population under healthy conditions. They reside immovably along the sinusoids, close to the entrance of supplying blood vessels to effectively sort out and process bypassing, gut-derived antigens and microbial products. Through close contact with parenchymal cells and the sinusoidal blood flow, Kupffer cells act as sensors of tissue integrity and gatekeepers for initiating or suppressing immune responses. In the adult liver of healthy mice, a minor proportion of macrophages derives from infiltrating monocytes (originating from the bone marrow). Upon liver injury, their number may rapidly expand. In mice, infiltrating Ly-6C^hi^ macrophages initiate inflammation while circulating Ly-6C^lo^ macrophages have a more mature phenotype, patrolling through the liver and promoting tissue restoration. After infiltration, Ly-6C^hi^ macrophages can mature to Ly-6C^lo^ macrophages. The peritoneal cavity contains GATA6^+^ macrophages that can promptly cross the mesothelium and infiltrate into the subcapsular area of the liver to support tissue repair. Typical markers of the different macrophage populations in mouse models are indicated.

**Figure 2 ijms-22-07249-f002:**
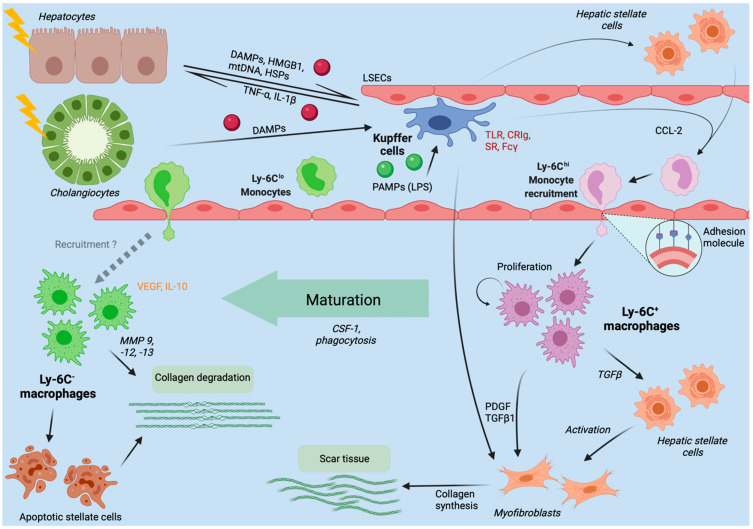
**Liver macrophages in the initiation, progression and regression of liver injury.** The figure summarizes the role of hepatic macrophages in the course of liver injuries, as derived from mouse models. Kupffer cells immovably reside at the luminal side of liver sinusoidal endothelial cells (LSECs). Upon injury to hepatocytes or cholangiocytes, intracellular components like mitochondrial DNA (mtDNA) or heat shock proteins (HSP), functioning as danger-associated molecular patterns (DAMPs), are released. Moreover, microbial products (e.g., LPS) from the intestinal microbiota enter the liver via the portal venous blood flow and function as pathogen-associated molecular patterns (PAMPs). DAMPs and PAMPs activate Kupffer cells, which in turn secrete inflammatory cytokines (e.g., TNF-α, IL-1β) and CCL-2. While TNF-α and IL-1β further contribute to hepatocyte injury, CCL-2 recruits Ly-6C^hi^ monocytes from the bloodstream to infiltrate the tissue and differentiate into inflammatory, fibrogenic and angiogenic Ly-6C^+^ macrophages. If the inflammation does not subside, Ly-6C^+^ macrophages will activate hepatic stellate cells (HSCs) to become collagen-producing myofibroblasts forming scare tissue. In case of inflammation resolution, Ly-6C^+^ macrophages will mature to restorative, anti-inflammatory Ly-6C^−^ macrophages, that promote degradation of scar tissue by matrix degrading metalloproteinases (MMPs).

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
