# Peer review of "Liver Injury and the Macrophage Issue: Molecular and Mechanistic Facts and Their Clinical Relevance"

_ijms, 2021, doi:10.3390/ijms22147249_

Round 1

Reviewer 1 Report

The authors summarized the role of hepatic macrophage based on the findings both of basic and clinical research.

The manuscript is very interesting; however, there are several minor issues for acceptance.

1. Line 306

The authors misunderstanding the criteria of NAFLD.

NAFLD is defined as alcohol intake: < 30 g/day for men, < 20 g/day for women (EASL–EASD–EASO. J Hepatol. 2016 Jun;64(6):1388-402.) or < 210 g/week for men, < 140 g/week for women (Chalasani N. Hepatology 2018 Jan;67(1):328-357).

2. Line 316

The authors misunderstanding the criteria of fibrosis stages.

Earlier fibrosis stage is defined as F0-F2, but not F0-F3. The authors should exchange “In earlier stage of NAFLD fibrosis (F0-F3)” to “In noncirrhotic NAFLD (F0-F3 fibrosis)”

3. Page 13

Galactin-3 and PPARs were described in 5.3. Therapeutic target: Macrophage polarization; however, I cannot understand whether these agents influence macrophage polarization. The authors should describe the brief function of these agents regarding macrophage polarization.

4. Line 580

Regarding belapectin, a subgroup analysis revealed that in patient without esophageal varices belapectin reduced HVPG and development varices, suggesting belapectin may be developed to reduce HVPG and prevent esophageal varices. The authors should mention these findings.

5. Line 186

“mt-DNA und nuclear DNA” is “mt-DNA and nuclear DNA”?

6. The abbreviation “non-alcoholic steatohepatitis (NASH)” was described several times. Line 90, 211, and 308.

Author Response

Dear reviewer,

thank you very much for your productive feedback. I have implemented the requested corrections. Also, I have changed the heading of the one section from "polarization of macrophages" to "function and polarization of macrophages" to include the effects of galactin 3 and PPAR agonism. 

You may find the corrected version attached,

Best

Siyer Roohani

Reviewer 2 Report

This is a well written and comprehensive review of Kupffer cell function.

No major concerns.

  1. Section 2, there are references to “embryonic week xx in mice” on two occasions, presumably this should be day not week? (since gestational period in mice is about 21 days) l(lines 70 and 71)
  2. There are inconsistencies with how NF-kB is written (NfkB, NFkB, NF-kB)
  3. There are also inconsistencies with how TNF-a is written (TNF-a and TNFa)
  4. There are also inconsistencies with how TGF-b is written (TGFb)
  5. Line 186, und instead of and
  6. line 224 should read “is rapidly expanded by” not “rapidly expands by”
  7. I think line 305 should read “ with no significant alcohol intake” instead of “particularly no significant alcohol intake”
  8. Line 269 guanine is spelled incorrectly

Author Response

Dear reviewer,

thank you very much for the productive feedback.

I have implanted the requested corrections. 
About correction number 7:

I used the phrase "particularly" intentionally to point out the definition of NAFLD being steatosis of no other causes and most importantly without significant alcohol intake ("Nonalcoholic fatty liver disease"). I have corrected the phrase.  

If you have further suggestions or corrections I am happy to implement them as well. 

Best regards,

Siyer Roohani
